# Retrospective Feature Estimation for Continual Learning

**Nghia D. Nguyen**[1,2]                                      *20nghia.nd@vinuni.edu.vn*

**Hieu Trung Nguyen**[3]                                      *thnguyen@se.cuhk.edu.hk*

**Ang Li**[4]                                                 *ang@simular.ai*

**Hoang Pham**[5]                                             *hoang.pham@warwick.ac.uk*

**Viet Anh Nguyen**[3]                                        *nguyen@se.cuhk.edu.hk*

**Khoa D Doan**[1]                                            *khoa.dd@vinuni.edu.vn*

[1] *VinUni-Illinois Smart Health Center*
[2] *University of Illinois Urbana-Champaign*
[3] *The Chinese University of Hong Kong*
[4] *Simular Research*
[5] *University of Warwick*

**Reviewed on OpenReview:** *https://openreview.net/forum?id=9NnhVME4Q6*

## Abstract

The intrinsic capability to continuously learn a changing data stream is a desideratum of deep neural networks (DNNs). However, current DNNs suffer from catastrophic forgetting, which interferes with remembering past knowledge. To mitigate this issue, existing Continual Learning (CL) approaches often retain exemplars for replay, regularize learning, or allocate dedicated capacity for new tasks. This paper investigates an unexplored direction for CL called Retrospective Feature Estimation (RFE). RFE learns to reverse feature changes by aligning the features from the current trained DNN backward to the feature space of the old task, where performing predictions is easier. This retrospective process utilizes a chain of small feature mapping networks called retrospector modules. Empirical experiments on several CL benchmarks, including CIFAR10, CIFAR100, and Tiny ImageNet, demonstrate the effectiveness and potential of this novel CL direction compared to existing representative CL methods, motivating further research into retrospective mechanisms as a principled alternative for mitigating catastrophic forgetting in CL. Code is available at: `https://github.com/mail-research/retrospective-feature-estimation`.

## 1 Introduction

Humans exhibit the innate ability to incrementally learn new concepts while consolidating acquired knowledge into long-term memories (Rasch & Born, 2007). More general Artificial Intelligence systems in real-world applications would require a similar imitation to capture the dynamics of the changing data stream. These systems need to acquire knowledge incrementally without retraining, which is computationally expensive and exhibits a large memory footprint (Rebuffi et al., 2016). However, existing learning approaches cannot match human learning in this so-called Continual Learning (CL) problem due to catastrophic forgetting (McCloskey & Cohen, 1989). These systems encounter difficulty in balancing the capability of incorporating new task knowledge while maintaining performance on learned tasks, or the plasticity-stability dilemma.

Representative CL approaches in the literature usually involve the use of a memory buffer for rehearsal (Ratcliff, 1990; Chaudhry et al., 2019a; Buzzega et al., 2020; Caccia et al., 2022; Bhat et al., 2023; Arani et al.,

2022; Prabhu et al., 2020), auxiliary loss term for learning regularization (Kirkpatrick et al., 2017; Ebrahimi et al., 2020; Zenke et al., 2017; Schwarz et al., 2018), or structural changes such as model pruning, growing or finding sub-networks (Rusu et al., 2016; Mallya & Lazebnik, 2018; Fernando et al., 2017; Yan et al., 2021; Serra et al., 2018; Wortsman et al., 2020; Kang et al., 2022). These methods share the common objective of discouraging the deviation of learned knowledge representation. Rehearsal-based methods allow the model to revisit past exemplars to reinforce previously learned representations. Alternatively, regularization-based methods prevent changes in parameter spaces by formulating additional loss terms. However, both approaches present shortcomings, including keeping a rehearsal buffer of all tasks during the model's lifetime or infusing ad hoc inductive bias into the regularization process. Meanwhile, structure-based methods utilize the over-parameterization property of the model by pruning, masking, adding parameters, or finding suitable sub-networks to reduce new task interferences.

This paper studies a novel approach for CL named Retrospective Feature Estimation (RFE), where we allow the model to "forget" knowledge of old tasks but then "correct" such "catastrophic forgetting" during inference using a sequence of lightweight feature mapping networks. These networks, called **retrospector** modules, help significantly reduce information loss in learned tasks by incrementally reversing changes in the feature space. Specifically, for each new task, we add a small, simple, and inexpensive auxiliary unit that aligns the feature from the current task to the previous task. Our method differs from many network expansion methods, in which additional parameters are allocated to minimize changes to the old parameters. Instead, extra modules are used to iteratively recover past representations by propagating the current representation backward through a series of mapping networks. With this mechanism, RFE allows the optimal learning of a new task (plasticity) while separately mitigating catastrophic forgetting through retrospectors (stability). RFE does not require saving past data to achieve strong performance, but can flexibly utilize past data to further improve capability. In addition, different from several CL approaches that heavily modify the training or network architecture, RFE imposes minimal changes to new task learning as modifications are mainly performed after the training has been completed using auxiliary modules. Hence, RFE can be easily integrated into existing CL pipelines.

**Contributions.** We propose a new paradigm for CL by sequentially correcting the current task's representation into the past task's representation using a chain of lightweight retrospector modules:

- We propose RFE, a novel approach to CL that separates catastrophic forgetting mitigation from new task learning via a sequence of lightweight retrospector modules. The proposed retrospector module, by compensating for information loss and reversing feature changes, can incrementally mitigate catastrophic forgetting.

- To train the retrospector modules, we rely only on task $t-1$'s feature extractor, and keeping past data is optional to improve performance. At inference time, for the task-incremental setting, we construct a chain of retrospector modules based on the provided task identity and forward the current features to correct the feature space. For the class incremental setting, RFE forms the final prediction from an average of predictions based on the reconstructed representations.

- We empirically evaluate our approach on three popular continual learning benchmarks (CIFAR10, CIFAR100, and Tiny ImageNet) to demonstrate that our approach achieves comparable performance with the existing representative CL directions.

This paper unfolds as follows. Section 2 discusses the literature on CL problems, and Section 3 describes the proposed RFE method. Finally, Section 4 provides the empirical evidence for the effectiveness of our proposed solution.

## 2 Related Work

Catastrophic forgetting is a critical concern in artificial intelligence and is arguably among the most prominent questions to address for DNNs. This phenomenon presents significant challenges when deploying models in different applications. Continual learning addresses this issue by enabling agents to learn throughout their

lifespans. This aspect has gained significant attention recently (Sun et al., 2022; Hu et al., 2021; Kirichenko et al., 2021; Balaji et al., 2020). Considering a model well-trained on past tasks, we risk overwriting its past knowledge by adapting it to new tasks. The problem of knowledge loss can be addressed using different methods, as explored in the literature (Yin et al., 2020; Farajtabar et al., 2020; Kirkpatrick et al., 2017; Li & Hoiem, 2018; Chaudhry et al., 2019a; Bhat et al., 2023; Rusu et al., 2016; Yan et al., 2021; Buzzega et al., 2020; Caccia et al., 2022; Arani et al., 2022; Prabhu et al., 2020; Ebrahimi et al., 2020; Zenke et al., 2017; Schwarz et al., 2018; Mallya & Lazebnik, 2018; Fernando et al., 2017; Serra et al., 2018; Wortsman et al., 2020; Kang et al., 2022). These methods aim to mitigate knowledge loss and improve task performance through three main approaches: (1) Rehearsal-based methods, which involve reminding the model of past knowledge by using selective exemplars; (2) Regularization-based methods, which penalize changes in past task knowledge through regularization techniques; (3) Parameter-isolation and Dynamic Architecture methods, which allocate subnetworks or expand new subnetworks, respectively, for each task, minimizing task interference and enabling the model to specialize for different tasks.

**Rehearsal-based.** Experience replay methods build and store a memory of the knowledge learned so far (Rebuffi et al., 2016; Lopez-Paz & Ranzato, 2017; Shin et al., 2017; Riemer et al., 2018; Rios & Itti, 2019; Zhang et al., 2019; Chaudhry et al., 2019a; Buzzega et al., 2020; Caccia et al., 2022; Bhat et al., 2023; Arani et al., 2022; Prabhu et al., 2020). As an example, Averaged Gradient Episodic Memory (A-GEM) (Chaudhry et al., 2019a) builds an episodic memory of parameter gradients, while DER (Buzzega et al., 2020) uses a reservoir sampling method to maintain episodic memory. These methods have shown strong performance in past studies, but they require a significant memory to store the examples.

**Regularization-based.** A popular early work using regularization is the elastic weight consolidation (EWC) method (Kirkpatrick et al., 2017). Other methods (Zenke et al., 2017; Aljundi et al., 2018; Van et al., 2022; Nguyen et al., 2018; Ahn et al., 2019; Ebrahimi et al., 2020) propose different criteria to measure the "importance" of parameters. A later study showed that many regularization-based methods are variations of Hessian optimization (Yin et al., 2020). These methods typically assume multiple optima in the updated loss landscape in the new data distribution. One can find a good optimum for both the new and old data distributions by constraining the deviation from the original model weights.

**Parameter Isolation.** Parameter isolation methods allocate different subsets of the parameters to each task (Rusu et al., 2016; Jerfel et al., 2019; Rao et al., 2019; Li et al., 2019; Serra et al., 2018; Kang et al., 2022). From the stability-plasticity perspective, these methods implement gating mechanisms that improve stability and control plasticity by activating different gates for each task. Masse et al. (2018) proposes a bio-inspired approach for a context-dependent gating that activates a non-overlapping subset of parameters for any specific task. Supermask in Superposition (Wortsman et al., 2020) is another parameter isolation method that starts with a randomly initialized, fixed base network and, for each task, finds a sub-network (supermask) such that the model achieves good performance.

**Dynamic Architecture.** Different from Parameter Isolation, which allocates subnets for tasks in a fixed main network, this approach dynamically expands the network structure. Yoon et al. (2018) proposes a method that leverages the network structure trained on previous tasks to effectively learn new tasks, while dynamically expanding its capacity by adding or duplicating neurons as needed. Other methods (Xu & Zhu, 2018; Qin et al., 2021) reformulate CL problems into reinforcement learning (RL) problems and leverage RL methods to determine when to expand the architecture when learning new tasks. Yan et al. (2021) introduces a two-stage learning method that first expands the previous frozen task feature representations by a new feature extractor, then re-trains the classifier with current and buffered data.

An orthogonal direction of CL is adapting a frozen pre-trained model to a sequence of tasks (for example, by learning new adapters), which share some similarities to our work but are not directly related. RFE assumes a more challenging scenario with training from scratch and evolving model parameters.

## 3 Proposed Framework

We consider the task-incremental learning (TIL) and class-incremental learning (CIL) scenarios, where we sequentially observe a set of tasks $t \in \{1, \ldots, N\}$. The neural network comprises a single task-agnostic

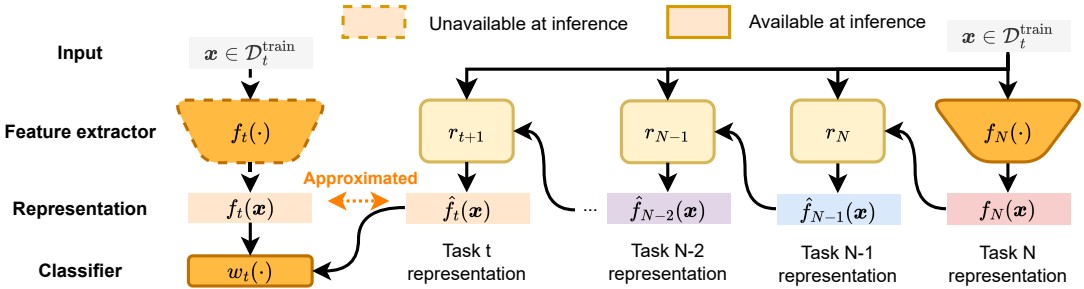

Figure 1: At task $t$, the feature extractor $f_t$ and classifier head $w_t$ are optimized on the dataset $\mathcal{D}_t^{\text{train}}$. During inference for a test sample from task $t$, we forward the input data $\boldsymbol{x} \in \mathcal{D}_t^{\text{test}}$ through the feature extractor and classifier head to obtain the logits. After learning all $N$ tasks, the DNN loses performance on task $t$ due to catastrophic forgetting. Therefore, the latent representation $f_N(\boldsymbol{x})$ is propagated through a series of retrospector module $r_N, \ldots, r_{t+1}$ to perform incremental latent rectification and obtained approximated representations $\hat{f}_{N-1}, \ldots, \hat{f}_t$. The logits can be obtained by passing the recovered representation to the respective classifier head.

feature extractor $f$ and a classifier $w$ with task-specific heads. The architecture of $f$ is fixed; however, its parameter set $\boldsymbol{\theta}_f$ is gradually updated as new tasks arrive. At task $t$, the system receives the training dataset $\mathcal{D}_t^{\text{train}}$ sampled from the data distribution $\mathcal{D}_t$ and learns the updated parameter sets $\boldsymbol{\theta}_f, \boldsymbol{\theta}_w$ of the feature extractor $f$ and $w$. To ease the discussion, the feature extractor and the classifier obtained after learning in task $t$ are denoted as $f_t$ and $w_t$, respectively. For an input-label pair $(\boldsymbol{x}, \boldsymbol{y})$ sampled from $\mathcal{D}$, the logits computed by $w$ is denoted as $\boldsymbol{z}$. Thus, after learning on task $t$, we obtain the evolved feature extractor $f_t$ and the classifier $w_t$. We call the latent space created by the feature extractor trained with $\mathcal{D}_t^{\text{train}}$ as the $t$-domain. Catastrophic forgetting occurs as the feature extractor $f_{t'}$ is updated into $f_t$ $(t' < t)$, which causes the $t'$-domain to be overwritten by the $t$-domain. This domain shift degrades the model's performance over time.

To overcome catastrophic forgetting, we propose a new CL paradigm: learning a retrospective feature estimation mechanism. This mechanism relies on a lightweight retrospector module $r_t$ that learns to align the features from the $t$-domain to the $(t-1)$-domain. Intuitively, this module "corrects" the feature change of a sample from the old task $t-1$ due to the evolution of the feature extractor $f$ when learning the newer task $t$. These retrospector modules will establish a chain of corrections for the features of any task's input, allowing the model to predict the past features better. Fig. 1 provides a visualization of the inference process on a task-$t$ sample, after learning $N$ tasks.

Learning the mechanism is central to our proposed framework. In general, each retrospector module should be small compared to the size of the final model or the feature extractor $f$, and its learning process should be resource efficient. The following sections present and describe our solution for learning this mechanism.

### 3.1 Learning the retrospector

As the training dataset $\mathcal{D}_t^{\text{train}}$ of task $t$ arrives, we first update the feature extractor $f_t$ and the classifier $w_t$. The primary goal herein is to find $(f_t, w_t)$ that has a high classification performance for task $t$, and the secondary goal is to choose $f_t$ that can reduce the catastrophic forgetting on previous tasks. To combat catastrophic forgetting, we will first discuss the objective function for learning the lightweight retrospector module $r_t$ and the potential options for training data.

### 3.1.1 Feature Estimation Loss

The goal of $r_t$ is to reduce the discrepancy between task $t$'s representation $f_t(\boldsymbol{x})$ and the $t-1$'s representation $f_{t-1}(\boldsymbol{x})$ for $\boldsymbol{x} \sim \mathcal{D}_{t-1}$; i.e., $r_t(f_t(\boldsymbol{x}), \boldsymbol{x}) \approx f_{t-1}(\boldsymbol{x})$. A simple choice is the $l_2$ error between $f_{t-1}(\boldsymbol{x})$ and $r_t(f_t(\boldsymbol{x}), \boldsymbol{x})$. Since the feature estimation loss will be reused multiple times in this paper, we define $s$ to be a function with parameter set $\boldsymbol{\theta}_s$ that encodes an input $\boldsymbol{x} \sim \mathcal{D}$ into its respective features. More specifically,

$s$ would serve as a placeholder for different functions in different training scenarios. We define the loss as:

$$\mathcal{L}_{\text{FE}}(\boldsymbol{\theta}_s; s, \mathcal{D}, f) = \mathbb{E}_{\boldsymbol{x} \sim \mathcal{D}} \left[ \|s(\boldsymbol{x}) - f(\boldsymbol{x})\|_2^2 \right] \tag{1}$$

For retrospector training, at task $t$, we set $s(\boldsymbol{x}) = r_t(f_t(\boldsymbol{x}), \boldsymbol{x})$, and aim to minimize the difference between $s$ and $f_{t-1}$; therefore, the objective function becomes

$$\mathcal{L}_{\text{FE}}(\boldsymbol{\theta}_{r_t}; r_t, \mathcal{D}, f_{t-1}) = \mathbb{E}_{\boldsymbol{x} \sim \mathcal{D}} \left[ \|r_t(f_t(\boldsymbol{x}), \boldsymbol{x}) - f_{t-1}(\boldsymbol{x})\|_2^2 \right] \tag{2}$$

### 3.1.2 Training Data

Training the retrospector $r_t$ to map the representation from the $t$-domain back to the $(t-1)$-domain requires the representations in both domains $(f_t(\boldsymbol{x}), f_{t-1}(\boldsymbol{x}))$ as training data. Ideally, the best choice would be to keep all the samples and the respective representations from task $t-1$ to train the retrospector module. However, it is impractical to keep all $\boldsymbol{x} \sim \mathcal{D}_{t-1}^{\text{train}}$ due to efficiency, scalability, or privacy issues. A practical approach is to only keep the previous task feature extractor $f_{t-1}$ and use current task samples $\boldsymbol{x} \sim \mathcal{D}_t^{\text{train}}$ to approximate the mapping of the representation space of the task $t$ back to that of the task $t-1$. Another approach is to keep only a subset $\mathcal{P} \subset \mathcal{D}_{t-1}^{\text{train}}$ of the previous tasks' samples and their respective representation. Nonetheless, the second approach heavily relies on the number of samples $|\mathcal{P}|$ that could be saved. Therefore, we opt for the first approach to train the retrospector. Keeping additional past data is optional and could be used to further improve performance. In this paper, we evaluate three strategies for training RFE.

**Without past task's data (RFE).** RFE can effectively recover $f_{t-1}(\boldsymbol{x})$ from $f_t(\boldsymbol{x})$ without relying on previous task's data. By only keeping the previous task feature extractor $f_{t-1}$ and using current task data $\boldsymbol{x} \sim \mathcal{D}_t^{\text{train}}$ as an approximation for $\mathcal{D}_{t-1}$, the retrospector module can learn to map from the $t$-domain to the $(t-1)$-domain. It is common for continual learning methods to exhibit performance degradation over time. However, even without access to any past task data, RFE demonstrates comparable performance to several rehearsal-based methods.

**With a subset $\mathcal{P}$ of task $t-1$'s samples (RFE-P).** In addition to keeping $f_{t-1}$, to improve rectification performance, a small subset of task $t-1$'s samples can also be saved together with their representation $f_{t-1}(\boldsymbol{x})$. With the use of additional past samples, RFE can sustain classification performance even under a long chain of retrospector modules. Therefore, RFE-P can be a good trade-off between performance and privacy since data are only stored with the maximum life cycle of 2 tasks (task $t-1$ and task $t$) rather than indefinitely as in buffer-based methods. However, it is possible to generalize this approach to store data from the past $k$ tasks, where $k = 1$ corresponds to RFE-P, meaning only task $t-1$'s samples are stored.

**With a buffer $\mathcal{B}$ of all tasks' samples (RFE-B).** Similar to many buffer-based methods, RFE can also make use of a reservoir buffer for a subset of all tasks' samples to sustain performance under very long retrospector module chaining. Instead of only being used for training the new retrospector module, these data samples can also be used to tune the learned retrospector modules, ensuring a stable rectification chain. On the other hand, unlike RFE and RFE-P, RFE-B can not be used in scenarios where privacy is a major concern.

Table 1: At task $t$, different training data require storing different components of the training process, which impose different trade-offs in terms of performance and privacy.

| Variation | Keep $f_{t-1}$ | Keep $\mathcal{P} \subseteq \mathcal{D}_{t-1}^{\text{train}}$ | Keep $\mathcal{B} \subseteq \cup_{i=1}^{t} \mathcal{D}_i^{\text{train}}$ |
|---|---|---|---|
| RFE | ✓ | - | - |
| RFE-P | ✓ | ✓ | - |
| RFE-B | ✓ | - | ✓ |

### 3.2 Retrospective Feature Estimation

The retrospective feature estimation mechanism relies on a chain of task-specific retrospector modules $(r_t)_{t=2}^{N}$ that aims to correct the distortion of the feature space as the extractor $f$ learns a new task.

### 3.2.1   Past Feature Estimation

For an input $\boldsymbol{x}$ at task $t-1$, its representation under the feature extractor $f_{t-1}$ is $f_{t-1}(\boldsymbol{x})$. One can heuristically define the $(t-1)$-domain as the representation space of the input under the feature extractor $f_{t-1}$. Unfortunately, the $(t-1)$-domain is brittle under extractor update: as the subsequent task $t$ arrives, the feature extractor is updated to $f_t$, and the corresponding features of the same input $\boldsymbol{x}$ will be shifted to $f_t(\boldsymbol{x})$. Likely, the $t$-domain and the $(t-1)$-domain do not coincide, and $f_t(\boldsymbol{x}) \neq f_{t-1}(\boldsymbol{x})$.

The retrospector module $r_t$ aims to offset this representation shift. To do this, $r_t$ takes $\boldsymbol{x}$, and its $t$-domain representation $f_t(\boldsymbol{x})$ as input, and the module outputs the approximated features that satisfies

$$r_t(f_t(\boldsymbol{x}), \boldsymbol{x}) \approx f_{t-1}(\boldsymbol{x}), \tag{3}$$

With this formulation, we can effectively minimize the difference between the rectified features $r_t(f_t(\boldsymbol{x}), \boldsymbol{x})$ and the original features $f_{t-1}(\boldsymbol{x})$.

### 3.2.2   Retrospector's Architecture

The proposed retrospector module comprises several trainable components: a weak *auxiliary feature extractor*, *soft gatings*, and *linear mappings*. The size of the retrospector module increases linearly with the number of tasks, similar to the classification head. However, since the retrospector module is lightweight, this is trivial compared to the size of the full model. Fig. 2 visualizes the module. The architecture of the retrospector is described here, with further implementation details included in the Appendix.

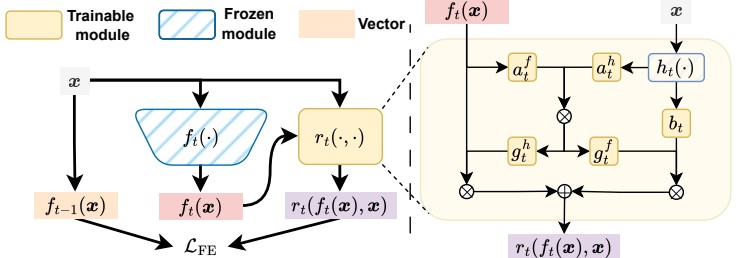

Figure 2: The retrospector module includes a weak auxiliary feature extractor $h_t$, linear mappings $a_t^f, a_t^h, b_t$, and soft gatings $g_t^f, g_t^h$. The joint information from the projected representations from both $f_t$ and $h_t$ is used to compute the gating value for the rectified representation.

**Auxiliary feature extractor $h_t$.** Due to catastrophic forgetting, the main feature extractor will gradually forget learned knowledge. Therefore, the auxiliary feature extractor will partially compensate for this loss of information. For our experiment, we chose a *simple and naive* design of an auxiliary feature extractor that has a low performance to demonstrate that the **effectiveness of RFE is based on retrospective feature estimation capability and not the auxiliary feature extractor's performance**. The auxiliary feature extractor $h_t$ processes the input data $x$ to generate a simplified representation $h_t(\boldsymbol{x})$. $h_t$ is distilled from $f_{t-1}$ to compress the knowledge of $f_{t-1}$ into a more compact, low-capacity parameter-efficient network. The weak auxiliary feature extractor is composed of only two 3x3 convolution layers and two max pooling layers. For simplicity and efficiency, instead of processing the full-size image, we use max-pooling to down-sample the input to 16x16 images before feeding it into $h_t$. The auxiliary feature extractor is a very small network compared to the main model.

**Linear mappings $a_t^f, a_t^h, b_t$.** To capture only relevant information and reduce noise from the main feature extractor and auxiliary feature extractor, both representations will be linearly projected down to a smaller-dimensional space $a_t^f \circ f_t(\boldsymbol{x})$ and $a_t^h \circ h_t(\boldsymbol{x})$, in which an element-wise multiplication $\odot$ is applied to combine both representation information.

$$a_t(\boldsymbol{x}, f_t(\boldsymbol{x}), h_t(\boldsymbol{x})) = \left(a_t^f \circ f_t(\boldsymbol{x})\right) \odot \left(a_t^h \circ h_t(\boldsymbol{x})\right) \tag{4}$$

with $\mathbf{dim}(a^f) = \mathbf{dim}(a^h) \ll \mathbf{dim}(f_t)$ ($\mathbf{dim}$ is the dimension of the output layer).

On the other hand, as $h_t$ may have a representation of a different dimension compared to that of $f_t$, another linear projection $b_t$ is used to map from $h_t$'s dimension to $f_t$'s dimension as

$$h_t'(\boldsymbol{x}) = b_t \circ h_t(\boldsymbol{x}) \tag{5}$$

**Soft gatings** $g_t^f, g_t^h$**.** To combine the main feature extractor and auxiliary feature extractor representations $f_t(\boldsymbol{x})$ and $h_t(\boldsymbol{x})$, an element-wise gating value $g_t^f(.)$ and $g_t^h(.)$ is computed from the encoded joint information $a_t(\boldsymbol{x}, f_t(\boldsymbol{x}), h_t(\boldsymbol{x}))$. As $g_t^f, g_t^h, a_t^f, a_t^h, b_t, h_t$ are components of the retrospector module, we compute the rectified representation as a function of the input $\boldsymbol{x}$ and its current representation $f_t(\boldsymbol{x})$ :

$$r_t(f_t(\boldsymbol{x}), \boldsymbol{x}) = \left( g_t^f \circ a_t(\boldsymbol{x}, f_t(\boldsymbol{x}), h_t(\boldsymbol{x})) \right) \odot f_t(\boldsymbol{x}) + \left( g_t^h \circ a_t(\boldsymbol{x}, f_t(\boldsymbol{x}), h_t(\boldsymbol{x})) \right) \odot h_t'(\boldsymbol{x})$$

The gating mechanism is simply a linear layer followed by sigmoid activation.

**Distinction from network-expansion approach.** It could be argued that one can, instead, separately train a weak feature extractor $h_t$ for each task, making it a network-expansion CL approach. However, because $h_t$ is a small and low-capacity network, this approach is ineffective; specifically, our experiments demonstrate that the task-incremental average accuracy across all tasks of this approach on CIFAR100 falls below 53%. Furthermore, for network expansion approaches, the dedicated parameters are allocated for new task learning, which fundamentally differs from RFE's objective to correct representation changes. The new task's knowledge is acquired by $f_t$ and $w_t$ with high plasticity.

### 3.3 Training Procedure

**Network training.** Similar to conventional DNN training, the performance of the feature extractor $f_t$ and the classifier head $w_t$ is measured by the standard multi-class cross-entropy loss:

$$\mathcal{L}_{\text{CE}}(\boldsymbol{\theta}_{f_t} \cup \boldsymbol{\theta}_{w_t}; w_t \circ f_t, \mathcal{D}_t^{\text{train}}) = \mathbb{E}_{(\boldsymbol{x}, \boldsymbol{y}) \sim \mathcal{D}_t^{\text{train}}} \left[ -\sum_{c=1}^{M_t} \boldsymbol{y}_c \log(\boldsymbol{z}_c) \right] \tag{6}$$

where $M_t$ is the number of classes of task $t$, $\boldsymbol{z}$ is the probability-valued network output for the input $\boldsymbol{x}$ that depends on the feature extractor $f_t$ and the classifier $w_t$ as $\boldsymbol{z} = w_t \circ f_t(\boldsymbol{x})$. Since the retrospector will correct feature changes back to the original feature space, classifier units that are learned from past tasks are excluded (masked) and not updated to prevent mismatch due to gradient updates.

Furthermore, to reduce forgetting and enable more effective rectification, we regularize (or distill) task $t-1$ representation knowledge by using the approximated previous task's representation from $f_{t-1}$ (and additional saved data in $\mathcal{P}$ or $\mathcal{B}$ if available) to train the current feature extractor $f_t$. Let $s(\boldsymbol{x}) = f_t(\boldsymbol{x})$, then we can similarly use $\mathcal{L}_{\text{FE}}$ in Eq. (1) with hyperparameter $\alpha$ in the training loss $\mathcal{L}_{\text{T}}$:

$$\mathcal{L}_{\text{T}}(\boldsymbol{\theta}_{f_t} \cup \boldsymbol{\theta}_{w_t}; \mathcal{D}_t^{\text{train}} \cup \mathcal{S}) = \mathcal{L}_{\text{CE}}(\boldsymbol{\theta}_{f_t} \cup \boldsymbol{\theta}_{w_t}; w_t \circ f_t, \mathcal{D}_t^{\text{train}}) + \alpha \mathcal{L}_{\text{FE}}(\boldsymbol{\theta}_{f_t}; f_t, \mathcal{D}_t^{\text{train}} \cup \mathcal{S}, f_{t-1}) \tag{7}$$

where $\mathcal{S}$ can be the empty set, the set $\mathcal{P}$, or the buffer $\mathcal{B}$ corresponding to RFE, RFE-P, or RFE-B, respectively.

**Retrospector training.** Training the retrospector module follows two main steps: train the weak auxiliary feature extractor $h_t$ at task $t-1$ and then the remaining components at task $t$. The weak feature extractor $h_t$ is distilled from $f_{t-1}$ as task $t-1$ training is completed using $\mathcal{L}_{\text{FE}}(\boldsymbol{\theta}_{h_t}; h_t, \mathcal{D}_{t-1}^{\text{train}}, f_{t-1})$ as in Eq. (1) with $s(\boldsymbol{x}) = h_t(\boldsymbol{x})$. After training, $h_t$ parameters are frozen to prevent modifications. Similarly, after the training of task $t$ is completed, we train the remaining components using $\mathcal{L}_{\text{FE}}(\boldsymbol{\theta}_{r_t} \backslash \boldsymbol{\theta}_{h_t}; r_t, \mathcal{D}_t^{\text{train}}, f_{t-1})$ as in Eq. (1) with $s(\boldsymbol{x}) = r_t(f_t(\boldsymbol{x}), \boldsymbol{x})$. For the case of RFE-B, the representations $f_t(\boldsymbol{x})$ of input $\boldsymbol{x} \in \mathcal{B}$ are rectified to their corresponding domains $i = \{1, 2, ..., t-1\}$ and learned retrospector's parameters $\cup_{i=1}^{t-1} \boldsymbol{\theta}_{r_i}$ are also optimized. Details of RFE's training algorithm are provided in Algorithm 1.

RFE imposes minimal changes to the standard training process as the majority of the additional training happens after the main (standard) training ends. However, end-to-end training of all modules can also be used as demonstrated in Section B.3.

### 3.4 Inference Procedure

We now describe how to stack multiple retrospector modules $r_t$ into a chain for inference. As a new task arrives, our model dynamically extends an additional retrospector module, forming a sequence of retrospector modules.

---

**Algorithm 1** Training process at task $t \in \{1, 2, ..., N\}$.

**let** $\mathcal{S}$ be $\emptyset$, $\mathbf{P}$, $\mathcal{B}$ for RFE , RFE-B, RFE-P, respectively
// main training starts
**optimize** $w_t \circ f_t$ with $\mathcal{L}_{\mathrm{T}}(\boldsymbol{\theta}_{f_t} \cup \boldsymbol{\theta}_{w_t}; \mathcal{D}_t^{\mathrm{train}} \cup \mathcal{S})$
// main training ends
**optimize** $h_{t+1}$ with $\mathcal{L}_{\mathrm{FE}}(\boldsymbol{\theta}_{h_{t+1}}; h_{t+1}, \mathcal{D}_t^{\mathrm{train}}, f_t)$
**freeze** $\boldsymbol{\theta}_{h_{t+1}}$
**if** $t > 1$ **then**
    **optimize** $r_t$ with $\mathcal{L}_{\mathrm{FE}}(\boldsymbol{\theta}_{r_t} \backslash \boldsymbol{\theta}_{h_t}, r_t, \mathcal{D}_t^{\mathrm{train}} \cup \mathcal{S}, f_{t-1})$
**end if**

---

Table 2: Task-Incremental Average Accuracy across all tasks after CL training. **Oracle**: the upper bound accuracy when jointly training on all tasks (i.e., multi-task learning). **Finetuning**: the lower bound accuracy when learning without CL techniques. $|\mathcal{B}|$ is the buffer of all past task samples. $|\mathcal{P}|$ is the subset of task $t-1$ training data. **params (training/inference)** is the number of parameters used during *training* (first value) and *inference* (second value) (lower is better), and *accuracy* is the average accuracy of all tasks (higher is better). For RFE, RFE-P, and RFE-B, parameters of all accumulated retrospectors are included for both training and inference.

| Method TIL | Exemplars | S-CIFAR10 | | S-CIFAR100 | | S-TinyImg | |
|---|---|---|---|---|---|---|---|
| | | params | accuracy | params | accuracy | params | accuracy |
| Oracle | - | 11.17/11.17 | $98.37_{\pm\, 0.12}$ | 11.22/11.22 | $86.57_{\pm\, 0.38}$ | 11.27/11.27 | $81.47_{\pm\, 0.22}$ |
| Finetuning | | | $60.08_{\pm\, 2.13}$ | | $24.90_{\pm\, 2.58}$ | | $13.67_{\pm\, 0.37}$ |
| AGEM | | 11.17/11.17 | $91.37_{\pm\, 0.40}$ | 11.22/11.22 | $65.50_{\pm\, 0.28}$ | 11.27/11.27 | $38.73_{\pm\, 1.23}$ |
| ER | | | $93.79_{\pm\, 0.96}$ | | $66.88_{\pm\, 0.50}$ | | $44.85_{\pm\, 0.99}$ |
| DER++ | | | $92.10_{\pm\, 0.74}$ | | $68.65_{\pm\, 0.93}$ | | $47.92_{\pm\, 0.73}$ |
| ER-ACE | $|\mathcal{B}|{=}500$ | | $93.60_{\pm\, 0.61}$ | | $67.97_{\pm\, 1.01}$ | | $47.88_{\pm\, 0.61}$ |
| ER-MKD | | 22.35/11.17 | $93.75_{\pm\, 0.39}$ | 22.44/11.22 | $70.63_{\pm\, 0.80}$ | 22.54/11.27 | $51.89_{\pm\, 0.24}$ |
| TAMIL | | 22.68/11.51 | $\underline{94.56}_{\pm\, 0.09}$ | 22.77/11.60 | $75.12_{\pm\, 0.25}$ | 23.20/12.03 | $63.28_{\pm\, 0.03}$ |
| CLS-ER | | 33.52/11.17 | $\mathbf{94.99}_{\pm\, \mathbf{0.25}}$ | 33.66/11.22 | $76.79_{\pm\, 0.47}$ | 33.81/11.27 | $50.28_{\pm\, 1.03}$ |
| RFE-P | $|\mathcal{P}| = 500$ | 23.76/12.59 | $92.94_{\pm\, 0.52}$ | 23.81/12.64 | $\underline{80.57}_{\pm\, 0.41}$ | 25.63/14.46 | $\mathbf{71.80}_{\pm\, \mathbf{0.51}}$ |
| RFE-B | $|\mathcal{B}| = 500$ | | $91.35_{\pm\, 0.30}$ | | $\mathbf{80.69}_{\pm\, \mathbf{0.41}}$ | | $\underline{69.91}_{\pm\, 0.36}$ |
| AGEM | | 11.17/11.17 | $90.26_{\pm\, 2.64}$ | 11.22/11.22 | $69.91_{\pm\, 0.62}$ | 11.27/11.27 | $45.58_{\pm\, 1.16}$ |
| ER | | | $94.91_{\pm\, 0.54}$ | | $72.17_{\pm\, 0.42}$ | | $53.98_{\pm\, 1.08}$ |
| DER++ | | | $93.35_{\pm\, 0.43}$ | | $72.90_{\pm\, 0.31}$ | | $57.17_{\pm\, 0.40}$ |
| ER-ACE | $|\mathcal{B}| = 1000$ | | $94.93_{\pm\, 0.40}$ | | $72.36_{\pm\, 0.68}$ | | $56.96_{\pm\, 0.51}$ |
| ER-MKD | | 22.35/11.17 | $\underline{95.28}_{\pm\, 0.04}$ | 22.44/11.22 | $74.04_{\pm\, 0.43}$ | 22.54/11.27 | $57.55_{\pm\, 0.53}$ |
| TAMIL | | 22.68/11.51 | $95.11_{\pm\, 0.31}$ | 22.77/11.60 | $77.94_{\pm\, 0.95}$ | 23.20/12.03 | $68.81_{\pm\, 0.85}$ |
| CLS-ER | | 33.52/11.17 | $\mathbf{96.02}_{\pm\, \mathbf{0.16}}$ | 33.66/11.22 | $79.82_{\pm\, 0.11}$ | 33.81/11.27 | $60.78_{\pm\, 0.40}$ |
| RFE-P | $|\mathcal{P}| = 1000$ | 23.76/12.59 | $92.92_{\pm\, 0.92}$ | 23.81/12.64 | $\underline{80.64}_{\pm\, 0.59}$ | 25.63/14.46 | $\mathbf{72.65}_{\pm\, \mathbf{0.56}}$ |
| RFE-B | $|\mathcal{B}| = 1000$ | | $90.74_{\pm\, 1.85}$ | | $\mathbf{81.06}_{\pm\, \mathbf{0.28}}$ | | $\underline{71.92}_{\pm\, 0.34}$ |
| RFE | - | 23.76/12.59 | $91.15_{\pm\, 0.11}$ | 23.81/12.64 | $79.54_{\pm\, 0.27}$ | 25.63/14.46 | $69.66_{\pm\, 0.17}$ |

**Task-Incremental.** We consider a task-incremental learning setting where a test sample $\boldsymbol{x}_i$ is coupled with a task identity $t_i \in \{1, \ldots, N\}$. To classify $\boldsymbol{x}$, we can recover $\hat{f}_{t_i}(\boldsymbol{x})$ by forwarding the current representation $f_N(\boldsymbol{x})$ through a chain of $N - t_i$ retrospector modules. We then pass this recovered latent variable through the classifier head $w_{t_i}$ to make a prediction. The output $\hat{\boldsymbol{y}}_i$ is computed as

$$\hat{\boldsymbol{y}}_i = w_{t_i}(\hat{f}_{t_i}(\boldsymbol{x})) \tag{8}$$

where $\hat{f}_{t_i}(\boldsymbol{x}) = r_{t_i+1}(\hat{f}_{t+i}(\boldsymbol{x}), \boldsymbol{x})$ with $t_i < N, \hat{f}_N = f_N$.

**Class-Incremental.** RFE relies on the task identity to reconstruct the appropriate sequence of retrospector modules for propagating the features to the original space. However, no identity is provided for the CL method in the class-incremental learning setting. We instead provided a simple method for inference without task identity, which demonstrates the method's extension to class-incremental learning; however, more robust task-identity inference methods could also be incorporated.

We obtain the class-incremental probabilities by forming an average of the class probabilities over all domains. For each domain, irrelevant classifier units (not belonging to the task) are excluded (masked) before computing the softmax probability. More specifically, from the current task $N$'s domain, we extract the representation $f_N(\boldsymbol{x})$ and iteratively rectify the latent back to task $N-1$, task $N-2$, ..., task 1's domain. These representations ($f_N(\boldsymbol{x})$ and $\hat{f}_{t_i}(\boldsymbol{x})|_{t_i=1}^{N-1}$) are then forwarded through respective classifier $w_{t_i}|_{t_i=1}^{N}$ to compute the softmax probabilities. We then average the softmax probabilities of all domains.

## 4 Experiments

Our implementation is based partially on the Mammoth (Boschini et al., 2022; Buzzega et al., 2020), TAMiL (Bhat et al., 2023), and CLS-ER (Arani et al., 2022) repositories.

### 4.1 Evaluation Protocol

**Datasets.** We select three standard continual learning benchmarks for our experiments: Sequential CIFAR10 (*S-CIFAR10*), Sequential CIFAR100 (*S-CIFAR100*), and Sequential Tiny ImageNet (*S-TinyImg*). Specifically, we divide S-CIFAR10 into 5 binary classification tasks, S-CIFAR100 into 5 tasks with 20 classes each, and S-TinyImg into 10 tasks with 20 classes each.

**Baselines.** We evaluate RFE against strong rehearsal-based CL methods, including ER (Chaudhry et al., 2019b), AGEM (Chaudhry et al., 2019a), DER++ (Buzzega et al., 2020), ER-ACE (Caccia et al., 2022), CLS-ER (Arani et al., 2022), TAMiL (Bhat et al., 2023), and ER-MKD Michel et al. (2024). We further provide an upper and lower bound for all methods by joint training on all tasks' data and fine-tuning without catastrophic forgetting mitigation. We employ ResNet18 (He et al., 2016) as the feature extractor for all benchmarks. The classifier comprises a fixed number of linear heads for each task.

Additional results and further details on datasets, baselines, and hyperparameters are provided in the supplementary materials.

Table 3: Class-Incremental Average Accuracy across all tasks after CL training. The settings are similar to Table 2.

| Method | Exemplars | S-CIFAR100 | S-TinyImg |
|---|---|---|---|
| Oracle Finetuning | - | $71.15_{\pm 0.68}$ $17.65_{\pm 0.10}$ | $58.23_{\pm 0.21}$ $7.73_{\pm 0.06}$ |
| AGEM | | $24.75_{\pm 0.98}$ | $9.30_{\pm 0.11}$ |
| ER | | $28.60_{\pm 0.66}$ | $10.09_{\pm 0.08}$ |
| DER++ | | $38.90_{\pm 1.20}$ | $13.50_{\pm 0.34}$ |
| ER-ACE | $|\mathcal{B}| = 500$ | $40.18_{\pm 0.80}$ | $17.36_{\pm 0.08}$ |
| ER-MKD | | $34.39_{\pm 1.16}$ | $12.64_{\pm 0.54}$ |
| TAMiL | | $44.43_{\pm 1.94}$ | $20.48_{\pm 0.55}$ |
| CLS-ER | | $\mathbf{50.68}_{\pm \mathbf{0.61}}$ | $20.25_{\pm 0.43}$ |
| RFE-P | $|\mathcal{P}| = 500$ | $45.53_{\pm 0.08}$ | $\mathbf{29.25}_{\pm \mathbf{0.56}}$ |
| RFE-B | $|\mathcal{B}| = 500$ | $\underline{45.85}_{\pm 0.56}$ | $\underline{27.29}_{\pm 0.50}$ |
| AGEM | | $26.66_{\pm 1.69}$ | $9.73_{\pm 0.30}$ |
| ER | | $34.77_{\pm 0.64}$ | $13.47_{\pm 0.68}$ |
| DER++ | | $45.92_{\pm 2.05}$ | $20.14_{\pm 0.80}$ |
| ER-ACE | $|\mathcal{B}| = 1000$ | $46.88_{\pm 0.86}$ | $22.96_{\pm 0.44}$ |
| ER-MKD | | $38.85_{\pm 0.90}$ | $16.09_{\pm 1.18}$ |
| TAMiL | | $\underline{50.37}_{\pm 1.00}$ | $28.21_{\pm 0.78}$ |
| CLS-ER | | $\mathbf{56.18}_{\pm \mathbf{0.20}}$ | $27.45_{\pm 0.69}$ |
| RFE-P | $|\mathcal{P}| = 1000$ | $45.85_{\pm 0.56}$ | $\mathbf{30.58}_{\pm \mathbf{0.28}}$ |
| RFE-B | $|\mathcal{B}| = 1000$ | $46.01_{\pm 0.54}$ | $\underline{30.47}_{\pm 0.51}$ |
| RFE | - | $43.63_{\pm 1.29}$ | $26.59_{\pm 0.44}$ |

### 4.2 Results

**Task-incremental.** Table 2 shows the performance of RFE, RFE-P, RFE-P, and other CL methods on multiple sequential datasets, including S-CIFAR10, S-CIFAR100, and S-TinyImg. From the table, we see that RFE-P and RFE-B achieve results comparable to the baselines on S-CIFAR10. On S-CIFAR100 and

S-TinyImg, RFE, without any past data, is equivalent to or outperforms all the baselines, including strong rehearsal-based methods such as TAMiL and CLS-ER, indicating its ability to estimate feature representation retrospectively. Notably, given either $\mathcal{P}$ or $\mathcal{B}$, RFE-P and RFE-B outperform the baselines.

**Class-incremental.** Table 3 demonstrates the extension of RFE-P and RFE-B to class-incremental settings. As the class-incremental probabilities are obtained through simple averaging, which may suffer from an overconfident classifier, we can achieve performance comparable to other methods on S-CIFAR100. On S-TinyImg, RFE-P and RFE-B demonstrate an improvement over other methods. The performance can be further optimized by integrating task identity prediction methods (i.e., OOD detection) Kim et al. (2022), which are left as potential improvements to avoid complicating the method.

**Long chaining.** Continual learning methods, including rehearsal-based approaches, often experience performance degradation over long task sequences. In Fig. 3, we demonstrate that RFE-P and RFE-B exhibit less forgetting than several continual learning methods across the ten tasks of S-TinyImg. In Fig. 4, we demonstrate the evolving average accuracies over 20 tasks of S-TinyImg, in which RFE is more stable and consistently improves over other methods.

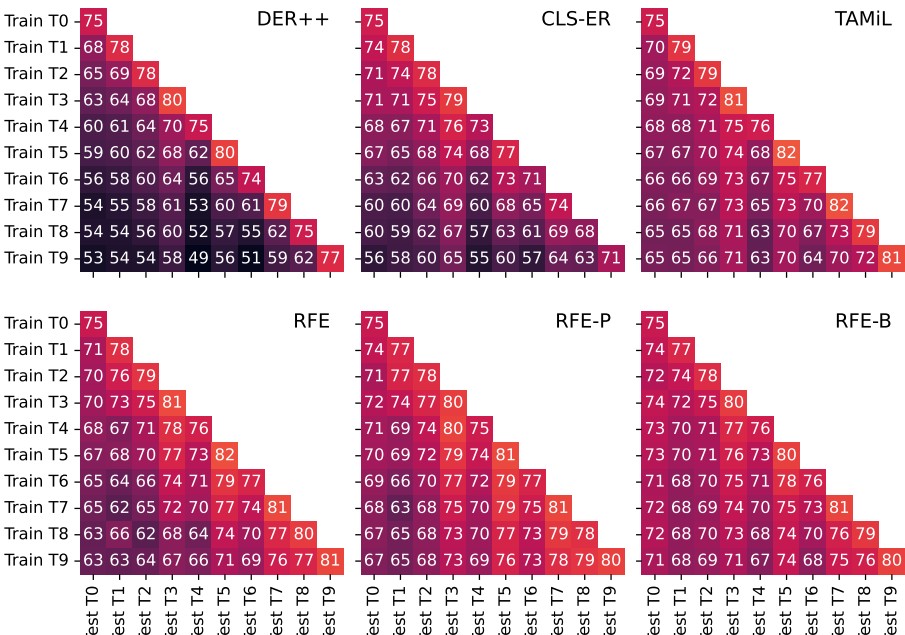

Figure 3: The TIL accuracy with 1000 exemplars on 10 tasks of S-TinyImg (lighter color is better). The vertical axis represents the task the model has been trained on. The horizontal axis represents the task identity. The value in the cell is the task's accuracy. RFE-P demonstrates a forgetting rate comparable to or better than other methods without revisiting distant task samples. RFE-B performance is more stable for long chaining.

**Overhead.** Further details on time and space overhead are in the supplementary materials, Sections B.1 and B.2.

**Other architectures.** Further results on ViT (Dosovitskiy et al., 2020) are in the supplementary materials, Section B.5. RFE demonstrates strong performance across both architectures.

### 4.3 Retrospector Experiment

While the retrospector has several components, the core idea is to efficiently combine $f(\mathbf{x})$ and $h(\mathbf{x})$ to rectify the representation using soft gating and linear layers for dimension mapping. Therefore, the components are tightly integrated and cannot be easily decomposed for individual testing. We cannot remove $h_t(\mathbf{x})$ as

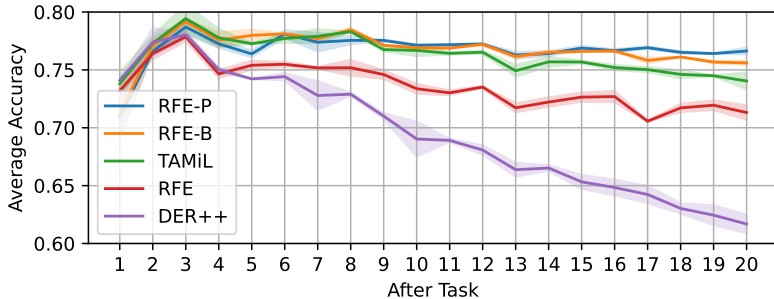

Figure 4: The evolving TIL average accuracies of CL methods with 1000 exemplars on 20 tasks of S-TinyImg. RFE-P and RFE-B consistently improve over baseline.

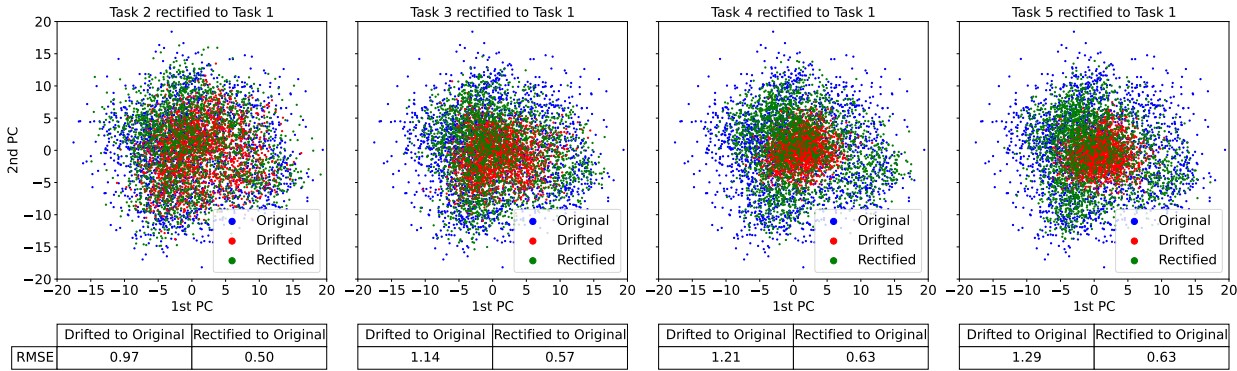

Figure 5: We employ PCA to visualize the rectified latent space after training on task $t$ and predicting task $t'(t' < t)$ of S-CIFAR100. By visualizing the original representation $(f_{t'}(\boldsymbol{x}))$, the drifted representation $(f_t(\boldsymbol{x}))$, the rectified representation $(\hat{f}_{t'}(\boldsymbol{x}))$, we demonstrate RFE effectiveness. The closer the rectified representation and the original representation, the better the performance. For Eq. (7), we set $\alpha = 0$ (no regularization) to clearly visualize the catastrophic forgetting and retrospector module performance.

we cannot recover $f_{t-1}$ with $f_t$ alone due to catastrophic forgetting. However, we can study the overall effectiveness of the retrospector by visualizing the representation of the "catastrophic forgetting" network and the rectified result by the retrospector. In Fig. 5, we utilize Principal Component Analysis (PCA) to visualize the latent space. The new representations of past data (red) after learning new tasks change significantly from the original representation (blue), which demonstrates catastrophic forgetting. With RFE, the rectified representations (green) align with the 'true' representations (blue), supporting the empirical effectiveness of our framework. The RMSE between representations is also computed. Additional ablation on alternative retrospector designs is in the supplementary materials, Section B.3.

## 5 Limitations

We have shown the potential and high utility of RFE's continual learning mechanism in this paper. Nevertheless, RFE also has some limitations. Despite being lightweight, RFE still maintains additional parameters, i.e., the retrospector module, which incurs an additional overhead as the number of tasks increases. Inference cost for a significantly long chain would be considerable, which can be improved with modified chaining methods such as skipping (i.e., building a retrospector every two tasks). RFE does not outperform state-of-the-art methods on the S-CIFAR10 dataset, which may indicate potential instability in rectifying fine-grained details of the representation. Additionally, since RFE relies on the task identity to reconstruct the retrospector sequence, application to class-incremental learning requires either inferring task identity or

averaging predictions. The current approach might suffer from overconfident classifiers. Class-incremental learning is still an open research area, where more effective adaptations of RFE can be discovered.

## 6 Conclusion

This work proposes a new CL paradigm. RFE tackles catastrophic forgetting through its novel Retrospective Feature Estimation mechanism that learns to align the newly learned representation of past data to their past representations. Unlike existing CL methods, RFE can operate as a data-free method while achieving comparable performance to rehearsal-based methods. Additional past data is optional and can be used to improve performance. Furthermore, RFE imposes minimal modification to task learning, as most of the training for rectification occurs after main task training.

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

# A    Detailed Experimental Setup

## A.1    Baselines

We evaluate RFE, RFE-P, RFE-B, ER, AGEM, DER++, ER-ACE, ER-MKD, CLS-ER, and TAMiL. More specifically, ER, AGEM, DER++, and ER-ACE are common rehearsal baselines that utilize a simple buffer. ER-MKD, TAMiL, and CLS-ER utilize additional exponential moving average (EMA) backbones for distillation. ER-MKD and TAMiL utilize 1 EMA backbone while CLS-ER utilizes 2 EMA backbones during training, which explains the high parameter usage in Table 2. TAMiL further uses task-specific auto-encoders during training and inference, which is also reflected in Table 2. RFE, RFE-P, RFE-B utilizes 1 additional backbone (training-only) and retrospectors (training and inference).

Due to the usage of additional EMA backbones and biology-inspired design, TAMiL and CLS-ER are SOTA rehearsal-based methods. Prior works that compare with CLS-ER and TAMiL often use the non-EMA version, while we use the stronger EMA version for fair comparison against our methods.

For comparison, we provide RFE-B and rehearsal-based methods with a buffer $\mathcal{B}$ with a max capacity of 500 and 1,000 samples. For RFE-P, we also provide a set $\mathcal{P}$ consisting of task $t-1$ data with 500 and 1,000 samples for fair evaluation.

For RFE-B, ER, DER++, ER-ACE, TAMiL, and CLS-ER, we employ the reservoir sampling strategy to remove the reliance on task boundaries as in the original implementation. On the other hand, RFE, RFE-P, RFE-B, AGEM, and TAMiL rely on the task boundary to learn the retrospector module, modify the buffer, or add a new task-attention module, respectively. For ER-MKD, we perform standard augmentation instead of multi-view augmentation. For TAMiL, we use the best-reported task-attention architecture. For CLS-ER, we perform inference using the stable backbone per the original formulation.

## A.2    Datasets

To demonstrate the effectiveness of our method, we perform empirical evaluations on three standard continual learning benchmarks: Sequential CIFAR10 (S-CIFAR10), Sequential CIFAR100 (S-CIFAR100), and Sequential Tiny ImageNet (S-TinyImg). The datasets are split into 5, 5, and 10 tasks containing 2, 20, and 20 classes, respectively. The dataset of S-CIFAR10 and S-CIFAR100 each includes 60,000 $32 \times 32$ images split into 50,000 training images and 10,000 test images, with each task occupying 10,000 training images and 2,000 testing images. The dataset S-TinyImg contains 110,000 $64 \times 64$ images with 100,000 training images and 10,000 test images divided into ten tasks with 10,000 training images and 1,000 test images each. We augment the data using random horizontal flips and random image cropping for each training and buffered image.

## A.3    Training

We employ ResNet18 (He et al., 2016) as the feature extractor for all methods and benchmarks. By default, the Resnet18's output dimension is $\mathbf{dim}(f) = 512$. For RFE's retrospector, we set $\mathbf{dim}(h) = \mathbf{dim}(a^f) = \mathbf{dim}(a^h) = 128$.

The training set of each task is divided into 90%-10% for training and validation. All methods are optimized by the Adam optimizer available in PyTorch with a learning rate of $5 \times 10^{-4}$. As the validation loss plateaus for three epochs, we reduce the learning rate by 0.1 times. Each task is trained for 40 epochs. For RFE, we train $h_t$ and $r_t$ using the same formulation with Adam optimizer at a learning rate of $5 \times 10^{-3}$ for 40 epochs.

## A.4    Hyperparameter search

For all methods, experiments, and datasets, we perform a grid search over the following hyperparameters in Table 4 using a validation set of 10% of the training data. Some hyperparameters are obtained directly from their original paper or implementation to narrow the search range. The final results are the average over 3 runs with the best hyperparameter, with different random seeds.

For RFE, a single value hyperparameter search is already sufficient in most cases.

Table 4: Hyperparameter search. See the original paper for each specific hyperparameter.

| Method | Hyperparameters Search range | Implementation name |
|---|---|---|
| Joint, Finetuning, ER, AGEM, ER-ACE | - | - |
| DER++ | $\alpha \in \{0.2, 0.5\}$ 
 $\beta \in \{0.5, 1.0\}$ | distill weight 
 replay weight |
| CLS-ER | $r_p \in \{0.5, 0.8\}$ 
 $r_s \in \{0.2, 0.5\}$ 
 $\alpha_p \in \{0.999\}$ 
 $\alpha_s \in \{0.999\}$ 
 $\lambda \in \{0.2, 0.5\}$ 
 $\gamma \in \{1.0\}$ | plastic frequency 
 stable frequency 
 plastic alpha 
 stable alpha 
 distill weight 
 replay weight |
| TAMiL | $\alpha \in \{0.5, 1\}$ 
 $\beta \in \{0.2, 0.5\}$ 
 $\lambda \in \{0.1\}$ 
 $\gamma \in \{0.05\}$ 
 $\eta \in \{0.999\}$ | replay weight 
 distill weight 
 pairwise weight 
 ema frequency 
 ema alpha |
| ER-MKD | $\lambda_\alpha \in \{2, 4\}$ 
 $\tau \in \{2, 4\}$ 
 $1 - \alpha \in \{0.99\}$ | distill weight 
 temperature 
 ema alpha |
| RFE, RFE-P, RFE-B | $\alpha \in \{1\}$ | regularize weight |

## A.5  Retrospector module

**Parameters.** The total number of parameters for each retrospector module is 0.35 million, with 0.08 million parameters occupied by the auxiliary feature extractor.

**Auxiliary feature extractor.** We provide the architecture of the auxiliary feature extractor $h_t$ in Table 5. We chose a simple design of two 3x3 convolution layers and two max-pooling layers. Depending on the use cases, a more robust feature extractor design can be used to improve performance and serve as a lower bound for the RFE. Nonetheless, to demonstrate that RFE depends on the rectification capability and not only the auxiliary feature extractor's performance, we opt to use a low-performance design.

Table 5: Architecture of the auxiliary feature extractor $h_t$. We use ReLU activation after each convolution.

| Layer | Channel | Kernel | Stride | Padding | Output size |
|---|---|---|---|---|---|
| Input | 3 | | | | $16 \times 16$ |
| Conv 1 | 64 | $3 \times 3$ | 2 | 1 | $8 \times 8$ |
| MaxPool | | | 2 | | $4 \times 4$ |
| Conv 2 | 128 | $3 \times 3$ | 2 | 1 | $2 \times 2$ |
| MaxPool | | | 2 | | $1 \times 1$ |

# B  Additional Experimental Results

## B.1  Time complexity

We report the training and inference time of the class-incremental setting on the S-TinyImg dataset in Table 6 to demonstrate the time overhead by the retrospector. For RFE, there is almost no overhead in training compared to ER, while for RFE-P and RFE-B, the training time moderately increased. For inference, the class-incremental setting represents the **worst-case** scenario for the RFE method, where features are propagated through all 10 tasks. Nonetheless, the inference time only moderately increased compared to other methods.

Table 6: Training and inference time for all 10 tasks on the S-TinyImg dataset with CIL setting.

|  | ER | TAMIL | CLSER | RFE | RFE-P | RFE-B |
|---|---|---|---|---|---|---|
| Training (hours) | 2.06 | 2.51 | 2.59 | 1.94 | 3.17 | 3.32 |
| Testing (second) | 7.36 | 7.42 | 7.71 |  | 11.54 |  |

## B.2  Space complexity

Combining the buffer size and the parameters in Table 2 in the main paper reflects the **memory footprint** of each method. **During training**, both the images and the params are loaded in the same `float32` format.

Consider S-TinyImg, for **buffer-only** methods (AGEM, ER, ER-ACE, DER++), we save 1000 $64 \times 64$ RGB images, which is approximately equivalent to 12.28 M params. RFE (no data) adds slightly more at 14.36 M params for the previous backbone and retrospectors. However, RFE's accuracy is 12.49%, higher than buffer-only methods (DER++).

Similarly, RFE-P and RFE-B additionally utilize saved data, while ER-MKD, TAMiL, and CLS-ER additionally utilize EMA backbone and/or auxiliary modules. Nonetheless, RFE-P outperforms TAMiL by 3.84% in average accuracy.

## B.3  Ablation on alternative designs

Table 7: TIL Average Accuracy across all tasks after CL training for RFE variants on S-CIFAR100.

| Type | Average Accuracy | Total Retrospectors' Parameters |
|---|---|---|
| RFE | $79.54_{\pm 0.27}$ | 1.42 |
| RFE end-to-end | $79.18_{\pm 0.44}$ | |
| MLP-Residual | $78.53_{\pm 0.62}$ | 2.66 |
| MLP-Projection | $52.78_{\pm 6.29}$ | 2.66 |

**Variants of retrospector**

A naive design is to map $f_t$ directly back to $f_{t-1}$ by directly using an MLP without using $h_t$. However, due to catastrophic forgetting, there is no straightforward method to recover information loss of task $t - 1$ without an external source ($h_t$).

A simple but inefficient alternative of the retrospector is to concatenate both $f_t(\boldsymbol{x})$ and $h_t(\boldsymbol{x})$, which are then forwarded through a simple MLP. We use a two-layer MLP with an output dimension of 512 for both layers and ReLU activation. The retrospector can then be used to learn the projection from task $t$ back to $t - 1$ (**MLP-Projection**):

$$r_t(f_t(\boldsymbol{x}), \boldsymbol{x}) = \mathbf{MLP}([f_t(\boldsymbol{x}), h_t(\boldsymbol{x})]) \tag{9}$$

or learn the residual of such projection (**MLP-Residual**):

$$r_t(f_t(\boldsymbol{x}), \boldsymbol{x}) = \mathbf{MLP}([f_t(\boldsymbol{x}), h_t(\boldsymbol{x})]) + f_t(\boldsymbol{x}) \tag{10}$$

Nonetheless, this design results in very high parameter usage while only delivering similar or worse performance than the gating design, as demonstrated in Table 7. The setting is similar to Table 2 in the main paper, using the S-CIFAR100 dataset.

**End-to-end training of RFE**

While RFE is designed to be mainly post-training to reduce interference with main network training, RFE can also function using end-to-end training as demonstrated in Table 7. In this case, post-training for $r_t$ and $h_{t+1}$ is no longer necessary. The main training loss $\mathcal{L}_\mathrm{T}$ in Algorithm 1 becomes:

$$\mathcal{L}_\mathrm{T}(\boldsymbol{\theta}_{f_t} \cup \boldsymbol{\theta}_{w_t}; \mathcal{D}_t^\mathrm{train} \cup \mathcal{S}) = \mathcal{L}_\mathrm{CE}(\boldsymbol{\theta}_{f_t} \cup \boldsymbol{\theta}_{w_t}; w_t \circ f_t, \mathcal{D}_t^\mathrm{train}) + \alpha \mathcal{L}_\mathrm{FE}(\boldsymbol{\theta}_{r_t}; r_t, \mathcal{D}_t^\mathrm{train} \cup \mathcal{S}, f_{t-1}) \tag{11}$$

## B.4 Feature estimation over a long sequence

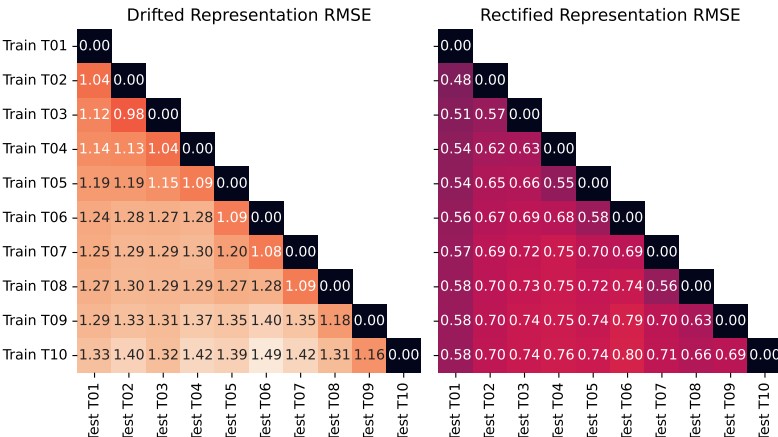

Figure 6: RMSE of the drifted representation and rectified representation against the original representation (darker color is better). Similar to Fig. 3, the vertical axis and horizontal axis represent the task the model has been trained on and the task identity, respectively. The retrospector is demonstrated to be effective in mapping from the drifted representation back to the original representation.

We extend the experiment in Fig. 5 to S-CIFAR100 10 tasks to demonstrate the effectiveness of the retrospector over longer sequences. Similarly, we set $\alpha = 0$ (no regularization) to clearly visualize the catastrophic forgetting and retrospector module performance. Each vertical column demonstrates the increasing forgetting of the model after training. Nonetheless, with the retrospector, we can effectively rectify the representation to reduce forgetting and maintain stable performance.

## B.5 Generalization to other architectures

We repeat a subset of the experiment for task-incremental learning with S-CIFAR100, but replacing the backbone from Resnet18 to a ViT-S/16 (Dosovitskiy et al., 2020) pre-trained on ImageNet-21k and fine-tuned on ImageNet-1k. The vision transformer backbone is followed by a linear layer with ReLU activation to ensure compatible dimensions with the ResNet18 backbone. Table 8 demonstrates the result of RFE using a vision transformer backbone to fine-tune 5 tasks of S-CIFAR100, which shares high similarity with results using a convolution neural network backbone. Each task is trained for 30 epochs with a learning rate of $10^{-5}$. Other training details are kept the same as in Section A.3.

Table 8: TIL Average Accuracy across all tasks after CL training using VIT-S/16 backbone.

| Method TIL | Exemplars | S-CIFAR100 | |
|---|---|---|---|
| | | params | accuracy |
| Oracle Finetuning | - | 21.91/21.91 | $96.93_{\pm 0.04}$ $78.88_{\pm 7.84}$ |
| AGEM ER DER++ ER-ACE | $\|\mathcal{B}\| = 500$ | 21.91/21.91 | $92.73_{\pm 0.15}$ $91.70_{\pm 0.62}$ $91.42_{\pm 0.33}$ $93.26_{\pm 1.26}$ |
| ER-MKD TAMiL CLS-ER | | 43.83/21.91 44.16/21.91 65.74/21.91 | $92.07_{\pm 0.88}$ $93.87_{\pm 0.23}$ $94.28_{\pm 0.28}$ |
| RFE-P RFE-B | $\|\mathcal{P}\| = 500$ $\|\mathcal{B}\| = 500$ | 45.19/23.28 | $\underline{95.58_{\pm 0.12}}$ $\mathbf{95.59_{\pm 0.18}}$ |
| RFE | - | 45.19/23.28 | $95.54_{\pm 0.01}$ |

## B.6 Comparison with non-rehearsal methods

RFE can be compared with existing works using Kim et al. (2022) and Bhat et al. (2023).

In Kim et al. (2022), the comparable setting is T-10T (similar to S-TinyImg). We consider the following methods: LwF (Li & Hoiem, 2018), HAT, (Serra et al., 2018), Sup (Wortsman et al., 2020), and HyperNet (von Oswald et al., 2020). It should be noted that we use 0 (RFE) or 1000 (RFE-P, RFE-B) samples with standard Resnet18, while methods in Kim et al. (2022) use 0 or 2000 samples with a larger Resnet18 that has double the channels. Consider the task-incremental learning setting in Table 7 of [4], the best baselines (both rehearsal and non-rehearsal) accuracy are 68.4, and the SOTA are HAT+CSI (72.4) and Sup+CSI (74.1). RFE (69.66) without data outperforms all baselines. RFE-P (72.65) and RFE-B (71.92) are in the same range of SOTA.

In Bhat et al. (2023), we consider PNN (Rusu et al., 2016), CPG (Hung et al., 2019), and PackNet (Mallya & Lazebnik, 2017). In Figure 2 of Bhat et al. (2023), TAMiL is demonstrated to outperform all baseline methods in task-incremental final average accuracy. In our experiments, RFE, RFE-B, and RFE-P outperform TAMiL in both CIFAR100 and Tiny ImageNet datasets for the task-incremental setting.

## C Versatility of RFE Framework

In RFE, as the tasks arrive, conventional fine-tuning or training on the new task happens with minimal CL's intervention. RFE only augments or adds to this process with a separate training of the retrospective feature estimation mechanism. The attractiveness of this framework is two-fold. First, RFE allows the best adaptation on the new task to possibly achieve maximum plasticity, while the backward rectification mechanism mitigates catastrophic forgetting. Second, unlike previous CL approaches that heavily modify the sequential training process, RFE minimally changes the fine-tuning process, allowing the users to flexibly incorporate this framework into their existing machine learning pipelines.

**Relationship to Memory Linking.** RFE's process of mapping newly learned knowledge representation resembles the popular human mnemonic memory-linking technique, which establishes associations of fragments of information to enhance memory retention or recall. [1] As the model learns a new task, the retrospector module establishes a mnemonic link from the new representation of the sample from the past task to its past task's correct representation.

---

[1] https://en.wikipedia.org/wiki/Mnemonic_link_system

