# OpenReview forum: "Retrospective Feature Estimation for Continual Learning"
_TMLR — Accepted by TMLR_

### Review · Reviewer_1diB · 2025-11-14

**Summary Of Contributions:**

This paper introduces a new approach to continual learning (CL) using deep neural networks (DNNs) termed Retrospective Feature Estimation (RFE). The main idea is to align the features learned by the DNN on the current task with features learned for previous tasks.

In more detail, the classifier for each task $t$ is decomposed into two parts: a feature extractor $f_t$ (mapping each data point to a set of learned features) and a classifier $w_t$ (mapping features to the logits for classification). The training enforces that features learned by $f_t$ should "encompass" the previous feature $f_{t-1}$. This is done by introducing a "retrospector module" $r_t$ that aims to recover $f_{t-1}(x)$ from $(f_t(x), x)$ for each training example $x$.

The method is evaluated on three continual learning datasets: the "sequential" versions of CIFAR10, CIFAR100, and Tiny ImageNet. Three variants of RFE are compared to several baselines in both task-incremental and class-incremental setups. The new methods give improved or comparable (within 5% of error) performance (Figures 1 and 2). The authors conducted additional experiments that visualize the degradation of performance over a long sequence of tasks (Figures 3 and 4) and the learned features (via PCA, in Figure 5).

**Audience:**

Yes

**Audience Explanation:**

This work proposes a new approach to continual learning, which, to the best of my knowledge, is quite different from previous ones. The method is very intuitive, nicely motivated, and shown to be effective. If the authors could give a convincing response to the questions above, I believe that this work would be interesting to the continual learning community.

**Broader Impact Concerns:**

None.

**Claims And Evidence:**

No

**Claims Explanation:**

Some aspects of the proposed method do not seem to be the most "natural" design choice, and they do not appear to be sufficiently motivated or justified in the manuscript. In addition, the PCA-based visualization in the experiments was a bit vague and unconvincing.

**End-to-end training?** Having defined the feature estimation loss (Equation (2)) and the architecture of the retrospector module (Section 3.2.2), arguably the most natural approach would be minimize the total loss (a weighted sum of the cross entropy and feature estimation losses) by training $f_t$, $w_t$, and $r_t$ (which includes $h_t$) simultaneously. Instead, the authors proposed a three-step training procedure (Section 3.3). Thus, I believe that some motivation / justification is in order, and it would be nice to have an ablation study that investigates the end-to-end alternative.

**The PCA-based visualization.** I found it a bit hard to understand Figure 5 and the discussion in Section 4.3. In the figures, the red points (the drifted representations) are much more concentrated around the origin compared to the blue ones (the original representations), while the green points (the rectified representations) are a lot closer to the blue ones. While this does suggest that the retrospector succeeds in matching the representations, I wasn't very convinced of the claim that the red-vs-blue comparison explains catastrophic forgetting---we could've obtained the red scatter from the blue one simply by scaling all features down by some factor, but this would keep all the information in the feature. Instead, using other metrics between feature spaces (e.g., the one of Wang et al. (NeurIPS'18)) would give a more convincing comparison.

**Effectiveness of feature estimation.** It would be enlightening if you evaluate the accuracy of the feature estimation module (the RMSE values reported in Figure 5) over a long task sequence. Specifically, if we store all the previous feature extrators $f_1, f_2, \ldots, f_{N-1}$ (for evaluation purpose only), for each $t \in [N-1]$, we can evaluate the distance between the actual representation $f_t(x)$ and the estimate $\hat f_t(x)$ (from Figure 1). It would be interesting to see how this error grows as we decrease $t$, which could say more about the effectiveness of the retrospector modules. (This could also be done by fixing $t = 1$ and increasing $N$, or having a 2D plot similar to Figure 3.)

**Reference**

Liwei Wang, Lunjia Hu, Jiayuan Gu, Zhiqiang Hu, Yue Wu, Kun He, John Hopcroft. Towards Understanding Learning Representations: To What Extent Do Different Neural Networks Learn the Same Representation. NeurIPS 2018.

**Requested Changes:**

Summarizing the above, the submission would be stronger if it included:
- Justification and/or motivation of the training procedure.
- An ablation study of an "end-to-end" variant of the training procedure.
- A more convincing interpretation on why Figure 5 "explains" catastrophic forgetting (or retraction of this claim).
- Optionally, a plot of the feature estimation error over a long task sequence.

Other comments/writing suggestions:
- Page 2: "the past task's representation" $\to$ "the past tasks' representation", if multiple past tasks are being referred to here.
- Page 3: There is a typo in Figure 1: "Available at infernece" $\to$ "Available at inference". Also, the legend for "Available at inference" is not very visible; consider making the outlines of the corresponding modules a bit thicker.
- Page 4: It might be clearer if you specify the "input" of $r_t$ when the notation is first introduced. Currently, it is not clear that $r_t$ takes $x$ (in addition to $f_t(x)$) as an input until Section 3.1.1.
- Page 6: The notations $\hat z$ and $\mathcal{L}_{\mathrm{align}}$ do not seem to be defined in the main text.
- Page 6: "will be linear projected down to" $\to$ "will be linearly projected down to"?
- Page 6: "a smaller dimension space" $\to$ "a smaller-dimensional space" or "a space of a lower dimension"?
- Page 6: Operations $\circ$ and $\odot$ should be defined when they first appear.
- Page 7: The use of set operations ("$\cup$" and "$\setminus$") on parameters might not be conventional; adding footnotes or a sentence at the first use could make things clearer.
- Page 7: The subscript "T" in the loss $\mathcal{L}_T$ was a bit unclear---does "T" stand for "Total"?
- Page 7: In Algorithm 1, the three for-loops over the epochs seem a bit redundant: the loop variables ($i$, $j$, and $k$) are not mentioned inside the loops. Removing those does not seem to affect readability, and you could save six lines of space.

---

### Review · Reviewer_68rz · 2025-11-25

**Summary Of Contributions:**

Summary Of Contributions:
This paper introduces Retrospective Feature Estimation (RFE), a new continual learning paradigm designed to mitigate catastrophic forgetting by rectifying the drift in feature representation after training on a new task and/or class. Instead of regularizing the model or extending the network, this method tries to rectifies the feature representation back into the feature space previously learned for that specific task.

Strengths:
- The idea of rectifying latent representations retrospectively is interesting and quite innovative. Rather than preventing drift, RFE adds lightweight modules to reverse drift, providing a clear conceptual distinction from typical rehearsal, regularization, or parameter-isolation approaches.
- RFE leaves standard training nearly untouched, avoiding complicating the primary learning objective of the network.
- The authors benchmark RFE across multiple datasets, task lengths, and CL scenarios (new tasks and classes).
- The PCA visualization clearly shows how retrospector modules recover past representations.

Weaknesses:
- Some aspects of the retrospector architecture are not extensively discussed. The authors might provide a more extensive discussion and/or ablation study on the several modules (auxiliary extractor, linear mappings, soft gatings) and their interplay.
- Even if each retrospector module is lightweight, inference requires chaining potentially many modules. Although the authors acknowledge this limitation, the paper lacks quantitative analysis and discussion of its practical implications for real-world deployment.

**Additional Comments:**

- Discussing failure cases and include examples where RFE does not fully recover past representations or where chaining becomes unstable, to clarify the practical limits of the method.
- Do you plan to share code and experiments?

**Audience:**

Yes

**Audience Explanation:**

The paper addresses one of the most central and longstanding challenges in continual learning—catastrophic forgetting, with a novel approach that does not rely on buffering, regularization, or dynamic architectures.

**Broader Impact Concerns:**

No concern.

**Claims And Evidence:**

Yes

**Claims Explanation:**

The authors show that their approach matches or improves over state-of-the-art methods across several datasets, reduces forgetting in long sequences, and accurately reconstructs past representations. The empirical evidence is consistent with the claims.

**Requested Changes:**

- Provide a more intuitive explanation of the gating + auxiliary extractor mechanism. Specify the dimensionality choices and justify them.
- The paper acknowledges inference-time cost but lacks detailed measurements. A table summarizing these aspects would help practitioners assess feasibility.
- Add a short subsection contrasting RFE with progressive networks and other expansion-based approaches, to better emphasize the conceptual novelty.
- While the technical novelty is appreciated, the paper would benefit from a clearer discussion of practical advantages. Why is RFE preferable to rehearsal-based methods or network expansion approaches in applied settings? Which applications would most benefit from this paradigm?

---

### Review · Reviewer_2PNf · 2025-11-26

**Summary Of Contributions:**

The paper proposes a novel Continual Learning (CL) framework called **Retrospective Feature Estimation (RFE)**. Instead of constraining the main model to prevent forgetting during training (which limits plasticity), RFE allows the model to freely learn new tasks and addresses forgetting via a post-hoc mechanism.

**Key contributions include:**
1.  **Methodology:** Introduction of "Retrospector" modules that learn to map/align the feature space of the current model back to the feature space of previous tasks. This effectively decouples the stability-plasticity dilemma: the backbone handles plasticity, while the retrospectors handle stability.
2.  **Privacy/Data Efficiency:** The method can function in a data-free (zero-shot) manner without storing old exemplars, addressing privacy concerns, though it can also leverage buffers (RFE-P/B) for improved performance.
3.  **Empirical Results:** The method demonstrates competitive performance on standard benchmarks (CIFAR-10, CIFAR-100, Tiny ImageNet) compared to established rehearsal-based methods, particularly in maintaining accuracy over longer task sequences.

**Strengths:**
* **Novelty:** The idea of "fixing" features via a chain of auxiliary modules is an interesting departure from standard weight regularization or replay methods.
* **Visualization:** The PCA visualizations (Fig. 5) provide convincing evidence that the retrospector effectively realigns drifted features.
* **Privacy:** Being able to operate without a replay buffer is a significant advantage for privacy-sensitive applications.

**Weaknesses:**
* **Inference Latency:** The sequential chaining of retrospector modules during inference introduces a linear increase in latency, which may be prohibitive for long task sequences.
* **Scalability:** While modules are lightweight, the parameter count still grows linearly with the number of tasks.
* **Class-Incremental Learning (CIL) Limitations:** The strategy for CIL (averaging predictions across all domain rectifications) seems computationally expensive and heuristic compared to the elegance of the Task-Incremental setting.

**Audience:**

Yes

**Audience Explanation:**

The paper addresses the fundamental problem of catastrophic forgetting in neural networks. The proposed perspective of "retrospective feature alignment" is conceptually distinct from the dominant replay/regularization paradigms. Researchers interested in Continual Learning, privacy-preserving ML, and dynamic network architectures would find this work relevant.

**Broader Impact Concerns:**

### **Broader Impact Concerns**

No concerns.

The paper discusses a fundamental learning algorithm. The authors also highlight the privacy benefits of their method (reducing the need to store raw user data), which is a positive ethical implication.

**Claims And Evidence:**

Yes

**Claims Explanation:**

The authors provide extensive experiments on standard CL benchmarks (S-CIFAR10, S-CIFAR100, S-TinyImg) and compare against strong baselines like DER++ and CLS-ER. The experimental setup is described in detail in the appendix. The ablation studies (e.g., comparing different retrospector designs like MLP-Projection vs. MLP-Residual) and PCA visualizations support the claim that the retrospector modules are indeed correcting feature drift rather than just acting as a generic ensemble.

**Requested Changes:**

I lean towards accepting this paper as it presents a novel and effective idea. However, there are several concerns regarding the practical deployment and analysis that the authors should address to strengthen the submission.

**Critical Adjustments:**

1.  **Inference Latency Analysis:**
    The paper briefly mentions overhead in Section 5 (Limitations), but it needs more quantitative data. In the Task-Incremental setting, inference for Task 1 after learning Task $N$ requires passing through $N-1$ retrospector modules.
    * **Request:** Please provide a plot or table showing the *inference time per sample* as a function of the number of tasks (e.g., from Task 1 to Task 20 on Tiny ImageNet). This will help readers understand the trade-off between accuracy and latency.

2.  **Clarification on CIL Strategy:**
    The approach for Class-Incremental Learning (CIL) involves "averaging predictions based on the constructed representations" across all domains. This implies that for every test sample, the model must perform inference $N$ times (one for each historical domain).
    * **Request:** Please explicitly discuss the computational cost of this strategy. Is it $N$ times slower than a standard classifier? The authors should acknowledge this computational bottleneck more clearly in the main text, as it significantly impacts the method's scalability compared to standard replay methods (e.g., ER/DER++) which use a fixed-size backbone.

3.  **Memory Overhead Clarification:**
    While "lightweight," the retrospector modules add parameters.
    * **Request:** In Table 2, please clarify if the "params" column includes the stored parameters of *all* retrospector modules accumulated up to that point. It is crucial to ensure the comparison is fair against dynamic architecture methods or fixed-capacity methods.

4.  **Scalability to Large Foundation Models (LLMs):**
The current experiments are limited to relatively small backbones (ResNet-18 and ViT-S/16). Given the increasing importance of Continual Learning in Large Language Models (LLMs), it would be valuable if the authors could discuss the scalability of RFE to billion-parameter models. Specifically: Training Memory Cost: Algorithm 1 implies that training the retrospector $r_t$ requires access to the previous frozen feature extractor $f_{t-1}$ to compute the loss. For LLMs (e.g., Llama-3-8B), keeping a copy of the previous model in memory alongside the current model for distillation/alignment is computationally expensive and VRAM-intensive.High-Dimensional Features: LLMs operate on much larger hidden state dimensions (e.g., 4096 or larger) compared to ResNet-18 (512). Would the "lightweight" retrospector need to be significantly larger to effectively capture the transformation in such high-dimensional spaces? Inference Latency in Autoregressive Generation: For LLMs, inference involves token-by-token generation. If RFE requires passing features through a chain of retrospectors for every token generated, the latency overhead might be prohibitive for real-time applications. A brief discussion on these potential bottlenecks would help position the paper better within the context of modern foundation models.

**Strengthening Suggestions (Non-critical):**

1.  **Comparison with "Expansion" Methods:** Since RFE adds parameters over time, it shares similarities with dynamic architecture/expansion methods (e.g., PNN, DYTOX). Including a brief discussion or comparison with a modern expansion-based method would better position the work.
2.  **Task ID Prediction:** The authors mention that OOD detection could be used for CIL. While not required, implementing a simple task-id predictor (even a baseline one) would make the CIL results much more convincing than the current "average all" heuristic.

---

### Decision · Action_Editor_CQiQ · 2026-01-05

**Recommendation:** Accept as is

**Additional Comments:**

Some minor comments for the camera ready version:
- While reading the revised paper I wondered why the retrospectors are not jointly trained with the feature extractors. I followed up the discussion with the reviewers and also had a look at the results. However, it might be worth mentioning the result (or pointing to the results in the appendix) at the end of Section 3.3
- Section 3.1 "we evaluate three strategies [to for] training" - sentence broken.
- Last sentence before Section 4: double ".."
- it might be beneficial to mention results for (or even applicability on) "bigger-than-CNN models" such as the Transformer already in the main text and linking to the results of the appendix. The discussion regarding bigger foundation models is indeed relevant but the reader yet does not see answers to this questions (unless by chance reading through the appendix).

**Audience:**

Yes

**Audience Explanation:**

The work addresses the key problem of catastrophic forgetting in continual learning and proposes a conceptually novel perspective, which is different from already known approaches (i.e, replay-, regularization-, or expansion-based paradigms): retrospective feature correction. All reviewers consistently noted that this reframing of the stability–plasticity dilemma is intuitive, well motivated, and complementary to existing approaches, particularly in settings where data storage is constrained by privacy or memory considerations. The empirical results across standard benchmarks, multiple continual learning scenarios, and both popular CNN and (vision-) transformer-based architectures further strengthen its relevance to researchers working on representation learning, continual and lifelong learning, and privacy-aware ML systems.

Hence, the findings presented in this paper are likely to reach a bigger audience within the TMLR community.

**Claims And Evidence:**

Yes

**Claims Explanation:**

The claims that this submission makes are supported by convincing and substantial evidence. One reviewer initially expressed concerns, particularly about the motivation of the staged training procedure and the interpretability of the PCA-based visualizations. But these issues have been addressed in the revision and the reviewer response. Across all reviewers, there is broad agreement that the empirical evaluation is thorough, including appropriate baselines, and spanning multiple datasets, task settings (TIL and CIL), and architectures (including Vision Transformers). Additional analyses on latency, parameter growth, and scalability trade-offs further align the experimental evidence with the paper’s claims and limitations.

The revised manuscript provides clear and credible empirical support for its main claims.